

# Repetitive low intensity magnetic field stimulation in a neuronal cell line: a metabolomics study

Ivan Hong[1], Andrew Garrett[2], Garth Maker[1], Ian Mullaney[1], Jennifer Rodger[2,3] and Sarah J. Etherington[1]

[1] School of Veterinary and Life Sciences, Murdoch University, Murdoch, WA, Australia
[2] School of Biological Sciences, Experimental and Regenerative Neuroscience, The University of Western Australia, Crawley, WA, Australia
[3] Brain Plasticity laboratory, Perron Institute for Neurological and Translational Science, Perth, WA, Australia

## ABSTRACT

Low intensity repetitive magnetic stimulation of neural tissue modulates neuronal excitability and has promising therapeutic potential in the treatment of neurological disorders. However, the underpinning cellular and biochemical mechanisms remain poorly understood. This study investigates the behavioural effects of low intensity repetitive magnetic stimulation (LI-rMS) at a cellular and biochemical level. We delivered LI-rMS (10 mT) at 1 Hz and 10 Hz to B50 rat neuroblastoma cells *in vitro* for 10 minutes and measured levels of selected metabolites immediately after stimulation. LI-rMS at both frequencies depleted selected tricarboxylic acid (TCA) cycle metabolites without affecting the main energy supplies. Furthermore, LI-rMS effects were frequency-specific with 1 Hz stimulation having stronger effects than 10 Hz. The observed depletion of metabolites suggested that higher spontaneous activity may have led to an increase in GABA release. Although the absence of organised neural circuits and other cellular contributors (e.g., excitatory neurons and glia) in the B50 cell line limits the degree to which our results can be extrapolated to the human brain, the changes we describe provide novel insights into how LI-rMS modulates neural tissue.

## INTRODUCTION

Faraday's discovery that a changing magnetic field induces a current in a conductor has contributed to many applications, such as the electromagnetic stimulation of body tissues and organs (*Barker, Jalinous & Freeston, 1985*). Specifically, electromagnetic stimulation of brain tissue has significant experimental and therapeutic potential because the induction of electric currents within neurons can modulate neuronal excitability, allowing non-invasive investigation and manipulation of brain circuit function and connectivity. Although clinical applications of magnetic fields most commonly involve high intensity fields that trigger action potentials and activate neural circuits (*Müller-Dahlhaus & Vlachos, 2013*), therapeutic effects are also observed during low intensity magnetic stimulation (*Di Lazzaro et al., 2013*; *Rohan et al., 2014*). For example in humans, low intensity repetitive

Corresponding author
Jennifer Rodger,
jennifer.rodger@uwa.edu.au

transcranial magnetic stimulation (LI-rTMS; pulse amplitude <100 mT) is beneficial in treating depression (*Martiny, Lunde & Bech, 2010*) and pain (*Shupak, Prato & Thomas, 2004*) and influences memory (*Navarro, Gomez-Perretta & Montes, 2016*).

It is apparent, however, that cellular mechanisms underpinning behavioural effects of LI-rTMS remain poorly characterised. Many studies in non-neuronal tissues and cells have focussed on the potentially negative effects of low intensity electromagnetic fields in the context of safety concerns surrounding the use of equipment emitting extremely low-frequency magnetic fields (ELF-MF; e.g., *ICNIRP, 1998*). However minimal investigation into low intensity stimulation in neuronal systems has been performed, despite demonstrated evidence of modulation of brain excitability in humans (*Capone et al., 2009*) and in animal models (*Yang, Ren & Mei, 2015*; *Balassa et al., 2013*). It has been established that the intensity of the induced electric fields in LI-rTMS is not sufficient to depolarise neurons to action potential firing threshold (*Davey & Riehl, 2006*; *Rudiak & Marg, 1994*). In contrast, *in vitro* experiments have consistently shown that low intensity repetitive magnetic stimulation (LI-rMS–no cranium) modulates intracellular calcium levels in non-neuronal (*Aldinucci et al., 2000*; *Walleczek & Budinger, 1992*; *Zhang et al., 2010*) and neuronal cells (*Grehl et al., 2015*).

We recently demonstrated that LI-rMS of dissociated cortical neurons rapidly increases levels of intracellular calcium (within 10 min of the onset of stimulation), with higher levels of intracellular calcium detected following 10 Hz compared to 1 Hz stimulation (*Grehl et al., 2015*). Such modulation of intracellular calcium alters NMDA receptor function (*Manikonda et al., 2007*) and provides a potential trigger for a wide range of changes in neuronal biochemistry which may underpin the LI-rTMS effects observed clinically (*Martiny, Lunde & Bech, 2010*; *Shupak, Prato & Thomas, 2004*). Further, there is also evidence that low intensity magnetic fields alter levels of biochemicals that function in neuronal processes, for example, low intensity magnetic fields modulate the level of the primary metabolite of serotonin, 5-HIAA, in rat brain in a dose (time)-dependent manner (*Shahbazi-Gahrouei et al., 2016*). In light of these findings, further investigation of biochemical and metabolic changes induced by LI-rMS in neuronal cells is warranted.

We hypothesize that changes in biochemical pathways due to LI-rTMS, will modify levels of a range of small molecule metabolites, including amino acids, carbohydrates and organic acids, which can be profiled using metabolomic techniques. Metabolomic analysis that profiles as many metabolites as possible in a single analysis is known as non-targeted screening. We performed such screening of a neuronal cell line immediately following 10 min of LI-rMS at 1 Hz or 10 Hz *in vitro*. We describe changes in the levels of 12 metabolites, 3 of which changed in a frequency-dependent manner.

## METHODS

### Cell culture

Rat neuroblastoma cells from the B50 cell line were seeded directly onto 6-well plates and grown for 24 h in media containing DMEM with 5% (v/v) heat-inactivated foetal calf serum, 2 mM L-glutamine, 100 U/ml streptomycin and 100 U/ml penicillin. Cells were

grown at 37 °C within a $CO_2$ incubator (5% $CO_2$ + 95% air). Cells from each 6-well plate were later pooled during extraction to make one replicate. Each stimulus condition or control had six replicates in total.

## LI-rMS stimulation

We used LI-rMS parameters that have previously been shown to increase intracellular calcium in primary cultured neurons *in vitro* (*Grehl et al., 2015*). Stimulation was delivered to cells in the incubator using custom built round coils (34 mm diameter, 17.1 mm height, 0.812 mm thickness, 138 turns). In order to deliver reproducible stimulation to each well, coils were designed to fit within a single well of a 6-well plate so that a plate containing cells could be placed on top of a plate containing five coils, resulting in reliable and reproducible placement at a distance of 2.8 mm from the base of each well (Figs. 1A and 1B). As the stimulator could only accommodate five coils, only five wells were stimulated on each plate. The coils were driven by a 12 V magnetic pulse generator under control of a programmable micro-controller card (CardLogix, Irvine, CA, USA), which delivered monophasic pulses (rise time of 0.725 ms). To approximate the stimulation dose in the 6-well plate, we measured the magnitude of the magnetic field in the $x$ and $y$ axes with a Hall Effect probe (SS94A2D; Honeywell, Morris Plains, NJ, USA). Magnetic field measurements were made at a distance of 0.5 cm from the base of the coil surface. Magnetic field strengths were converted to dB/dT, as a measure of the change in magnetic field over time of each delivered pulse. A map of dB/dT across the 6-well plate was generated with Matlab (Fig. 1C). Peak magnetic field and dB/dT inside each well ranged between ∼2.2 and ∼2.4 mT and ∼3 to ∼3.2 T/s respectively. Coil temperature did not rise above 37 °C, ruling out confounding effects of temperature change. Vibration of the coil was assumed to be within vibration amplitude of background (bench surface), as shown previously in *Grehl et al. (2015)*. For the experiments, 6-well plates were assigned to one of three conditions: control (pulse generator switched off, no LI-rMS) ($n = 6$), 1 Hz (continuous stimulation for 10 min at 1 Hz) ($n = 4$) and 10 Hz (continuous stimulation for 10 min at 10 Hz) ($n = 6$). Stimulation and sham (control) stimulations were performed at 37 °C within the $CO_2$ incubator.

## Metabolite extraction and derivatisation

Cells were immediately quenched and washed twice post-stimulation with 2 ml of ice-cold phosphate-buffered saline. Cells were then scraped, aggregated and frozen in liquid nitrogen before being freeze-dried. 500 μl of extraction solution consisting of 2.6 μg/ml of $^{13}C_6$-sorbitol, as an internal standard, was dissolved in methanol and added to the dried cells. Cells were lysed with a Precellys 24 tissue lyser (Bertin Technologies, Saint-Quentin en Yveline, France) at 6,500 rpm for 40 s. The supernatant was evaporated before adding 500 μl of water and freeze-drying again. Samples were derivatised according to the protocol described in *Abbiss et al. (2012)*. 20 μl of methoxyamine HCl solution (20 mg/ml in pyridine) was added to each extract and agitated at 1,200 rpm and 30 °C for 90 min. 40 μl of N-methyl-N-(trimethylsilyl) trifluoroacetamide (MSTFA) was subsequently added to each extract and agitated at 300 rpm and 37 °C for 30 min. The derivatised samples were transferred to GC vials and 5 μl of alkanes($C_{10}$, $C_{12}$, $C_{15}$, $C_{19}$, $C_{22}$, $C_{28}$; 0.156 mg mL$^{-1}$;

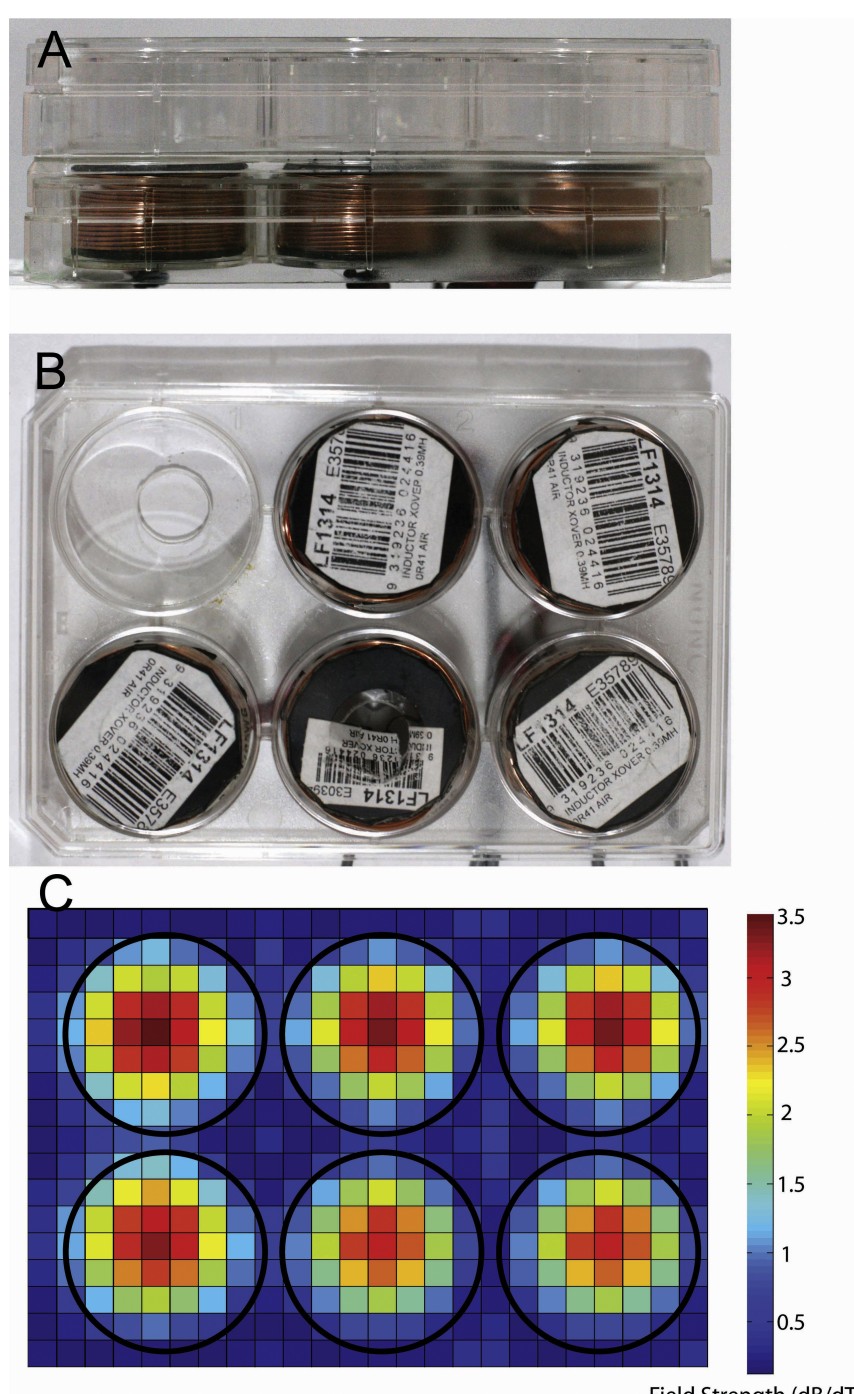

**Figure 1  Photographs showing equipment used for magnetic stimulation of cells.** Stimulation equipment. (A, B) Photographs of the *in vitro* stimulation coils used in this study. Views are from the side (A) and top (B). The coils were situated at a distance of 2.8 mm from the bottom of the culture well because of the thickness of the coverplate and base of the culture dish. (C) Heatmap showing the measured change in magnetic field for a single pulse (dB/dT) when 6 coils are placed in a 6-well plate arrangement. Note the lack of overlap of magnetic field between the coils/wells of the plates.

$C_{32}$, $C_{36}$; 0.313 mg/ml) in hexane were added to each sample for calculation of a Kovat's retention index, which aids comparison of data between samples.

## Sample analysis

An Agilent 6890 series gas chromatograph with an Agilent 7863 autosampler coupled to an Agilent 5973N single quadrupole gas chromatography-mass spectrometer(GC-MS) (Agilent Technologies, Australia, Mulgrave, Australia) was used along with a Varian Factor Four-fused silica capillary column VF-5ms (30 m × 0.25 mm × 0.25 µm + 10 m EZ-Guard). The analytical method was also as described in *Abbiss et al. (2012)*. 1 µl of the sample was injected splitless into the inlet that was set to 230 °C. The oven temperature was initially set to 70 °C with an initial temperature ramp of 1 °C/min for 5 min and was subsequently set to 5.63 °C/min, to a final temperature of 330 °C and held for 10 min. The ion source was set to 70 eV and 230 °C while the transfer line to the mass spectrometer was set to 330 °C. The detector, set to full scan, monitored a mass range of m/z 45–600 at 1 scan per second. The carrier gas, helium, was set at a flow rate of 1 ml/min.

## Data analysis

Due to co-elution, the peaks of the $^{13}C_6$-sorbitol internal standard and glutamate could not be deconvoluted. Results were therefore normalized using the total ion chromatogram (TIC). The presence of aberrant peaks suggested that two of the 6-well plates (1 Hz stimulation condition) were not adequately washed during the extraction process and these were therefore excluded from the final analysis.

   GC-MS data were viewed with AnalyzerPro v2.70 (SpectralWorks, Runcorn, UK). The mass spectra of peaks from the chromatogram were matched against the NIST (National Institute of Standards and Technology) mass spectral library. Metabolites with a similarity index of more than 60% were tentatively identified as metabolites. Metabolites with multiple derivative peaks were summed and treated as a single metabolite. For the peak area matrix, features that occurred in less than 80% of all samples were removed, unless they were unique to one treatment group in >75% of replicates. Peak areas were normalised to the total ion chromatogram (TIC) and imported into Unscrambler X version 10.1 (CAMO Software, Oslo, Norway). The data was log transformed $[X = \log(x + 1)]$ and a principal component analysis (PCA) was performed using a non-iterative partial least squares algorithm, cross validation and no rotation. Statistical comparisons between treatment samples were conducted with SPSS v21 (IBM, Corporation, Armonk, NY, USA) using a one-way ANOVA with Tukey's post-hoc test.

## RESULTS

Based on GC-MS profiling, a total of 18 reproducible intracellular metabolites were identified from the total ion chromatograms (TIC) (Table 1). At 1 Hz, significant decreases were observed in a total of 12 metabolites. This included seven amino acids, namely alanine, glycine, isoleucine, phenylalanine, serine, threonine (all $p < 0.01$) and aspartate (all $p < 0.05$). Significant decreases were also observed in cholesterol, glycylglutamic acid, inositol, pyroglutamate and succinate (all $p < 0.01$). Non-significant decreases were observed in three carbohydrates: fructose, galactose and glucose.

**Table 1 Fold change of metabolites following magnetic stimulation.** Intracellular metabolites identified by GC-MS and PCA as contributing the most to the variance between unstimulated controls and cells stimulated at 1 or 10 Hz and the fold change observed between stimulated cells and controls 1 Hz ($n = 4$), 10 Hz ($n = 6$) and unstimulated controls ($n = 6$).

| Metabolite | Fold change (1 Hz) | Fold change (10 Hz) | F-statistic |
|---|---|---|---|
| **Amino acids** | | | |
| Alanine | 0.625[**] | 0.873 | 6.477 |
| Aspartate | 0.322[*] | 0.619 | 3.846 |
| Glycine | 0.570[**] | 0.761[**] | 19.162 |
| Isoleucine | 0.599[**] | 0.753[*] | 7.835 |
| Phenylalanine | 0.379[**] | 0.477[**] | 10.710 |
| Serine | 0.389[**] | 0.656[*] | 15.784 |
| Threonine | 0.606[**] | 0.764[*] | 11.286 |
| Valine | 0.942 | 0.956 | 0.064 |
| **Carbohydrates** | | | |
| Fructose | 0.638 | 0.958 | 1.788 |
| Galactose | 0.629 | 0.948 | 1.149 |
| Glucose | 0.858 | 0.938 | 0.161 |
| **Other metabolites** | | | |
| Carbonic acid, 4-methylphenyl phenyl ester | 1.268 | 1.267 | 2.021 |
| Cholesterol | 0.713[**] | 0.831[*] | 8.909 |
| Glycerol-3-phosphate | 0.801 | 0.798 | 4.155 |
| Glycylglutamic acid | 0.526[**] | 0.693[**] | 26.740 |
| Inositol | 0.563[**] | 0.742 | 7.881 |
| Pyroglutamate | 0.531[**] | 0.727[*] | 9.603 |
| Succinate | 0.640[**] | 0.803[*] | 11.461 |

Notes.
Statistical significance was determined using a one-way ANOVA with Tukey's post-hoc test and is indicated as *, $p \leq 0.05$; **, $p \leq 0.01$.

At 10 Hz, significant decreases were observed in a total of nine metabolites, all of which were also significantly decreased at 1 Hz. These were the amino acids glycine, phenylalanine (both $p < 0.01$), isoleucine, serine and threonine (all $p < 0.05$), as well as glycylglutamic acid ($p < 0.01$), cholesterol, pyroglutamate and succinate (all $p < 0.05$). Non-significant decreases were also observed in the same three carbohydrates as for 1 Hz. Measured as a fold change, the decrease observed at 1 Hz was greater than at 10 Hz for all 12 significantly different metabolites.

In order to visualise the metabolomic data, relative peak areas (as measured by GC-MS) were subjected to principal component analysis (PCA). The PCA scores plot could be separated into three groups based on the frequency at which the cells were stimulated (Fig. 2). The majority of the variation between control, 1 Hz and 10 Hz could be attributed to principal component 1 (PC-1), which accounted for 63% of the variance.

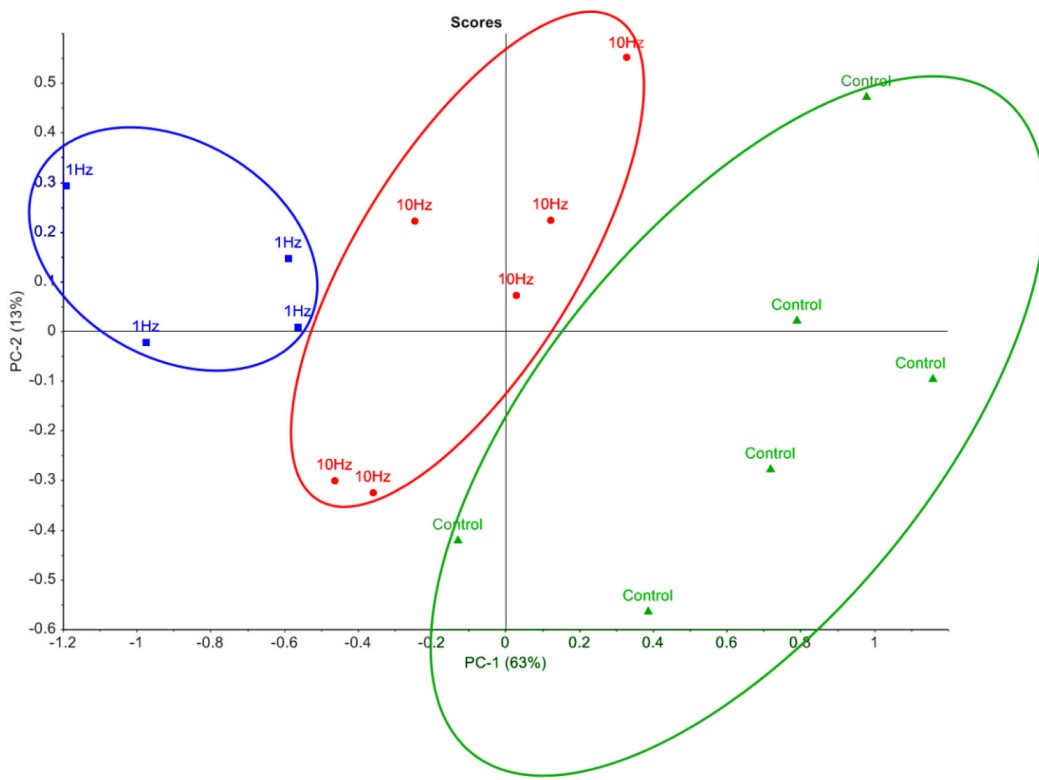

**Figure 2** **Principal component analysis of metabolic profiles.** Principal component analysis score plot of intracellular metabolite profiles (each data point represents a replicate sample) from GC-MS analysis of cells stimulated with LI-rMS at 1 Hz ($n = 4$), 10 Hz ($n = 6$) and unstimulated controls ($n = 6$).

## DISCUSSION

This study demonstrates for the first time that *in vitro* application of LI-rTMS depletes selected metabolites in B50 cells, including numerous amino acids. Furthermore, we confirm that LI-rMS effects are frequency-specific with 1 Hz stimulation having stronger effects than 10 Hz.

### LI-rTMS induces specific metabolic changes associated with GABA synthesis

Following LI-rTMS, we did not detect any significant change in the levels of glucose, fructose or galactose, which are key energy supplies to the cell. This is in contrast with outcomes of high intensity rTMS in human and in cat studies, where changes in glucose levels were detected during and immediately after stimulation (*Valero-Cabré, Payne & Pascual-Leone, 2007*). In the latter studies, glucose uptake was significantly reduced during stimulation, suggesting a local suppression of neuronal firing (*Valero-Cabré, Payne & Pascual-Leone, 2007*). However, immediately after high frequency, high intensity rTMS, the same regions had increased metabolism (*Valero-Cabré et al., 2005*), presumably due to long lasting plastic changes in intracortical circuitry (*Walsh & Pascual-Leone, 2003*).

Together, these findings suggest that the intensity and frequency of stimulation contribute to changes in metabolic response.

However, in contrast to the stability of major carbohydrates, the metabolic profiles of metabolites implicated in GABA synthesis were reduced following LI-rMS. GABA is synthesised via glutamate, from α-ketoglutarate, a principal component of the TCA cycle (Fig. 3, box a). We observed depletion of three amino acids involved in *de novo* synthesis of TCA cycle intermediates, aspartate, phenylalanine and isoleucine (see Fig. 3, boxes b, c and d, respectively). A possible explanation is that LI-rMS caused increased spontaneous neurotransmitter release in B50 cells, requiring the use of α-ketoglutarate to replenish GABA pools and depleting the downstream TCA cycle intermediates and their substrates (Table 1 and yellow boxes, Fig. 3). Similarly, increased GABA synthesis could explain the LI-rTMS dependent reduction in pyroglutamate and alanine (*Kumar & Bachhawat, 2012*; *Westergaard et al., 1993*), both of which can be converted to glutamate and subsequently to GABA. In support of this, spontaneous transmitter release is increased following LI-rMS in excitatory neurons (*Ahmed & Wieraszko, 2008*), and our results suggest a similar increase in spontaneous GABA release from B50 cells.

## Reduction in other amino acids and metabolites

The amino acids serine and glycine were also significantly decreased following 1 Hz and 10 Hz LI-rMS. The technique used in our study could not differentiate between L- and D-serine. Although both stereoisomers are found in the brain, only L-serine is incorporated into proteins, while D-serine acts as a neuromodulator by co-activating NMDA receptors (*Wolosker, 2006*). The reduction in serine could be due to increased protein synthesis, which has been shown for specific proteins such as BDNF, c-fos and various neurotransmitter receptors following rTMS (*Chervyakov et al., 2015*; *Rodger et al., 2012*). Glycine is mostly synthesized *de novo* in the brain from serine, rather than taken up via the blood–brain barrier (*Shank & Aprison, 1970*), and accumulation of glycine in neurons requires the activity of the glycine transporter GlyT2 (*Gomeza et al., 2003*). Thus, disruption of L-serine transport and/or diminished conversion of L-serine into glycine may contribute to the observed reduction in glycine. A key function for serine and glycine is to act as co-agonists for the NMDA receptor, which plays a central role in the long term plastic changes induced by high intensity rTMS(*Vlachos et al., 2012*), and may also contribute to LI-rTMS effects (*Makowiecki et al., 2014*; *Rodger et al., 2012*).

## LI-rMS downregulates inositol and cholesterol—implications for calcium signalling and exocytosis

Inositol plays a key role as a precursor for inositol lipid and inositol phosphate syntheses, which are vital for signal transduction and intracellular calcium homeostasis (*Wen, Osborne & Meunier, 2011*). A previous study showed that LI-rMS increases the levels of intracellular calcium in cortical neurons by release from intracellular stores, possibly via inositol-1,4,5-triphosphate (IP3) signalling (*Grehl et al., 2015*). Inositol depletion as detected here may thus reflect the changes in intracellular signalling events induced by LI-rMS. Cholesterol was also decreased in our study. This lipid is not only important as a structural constituent

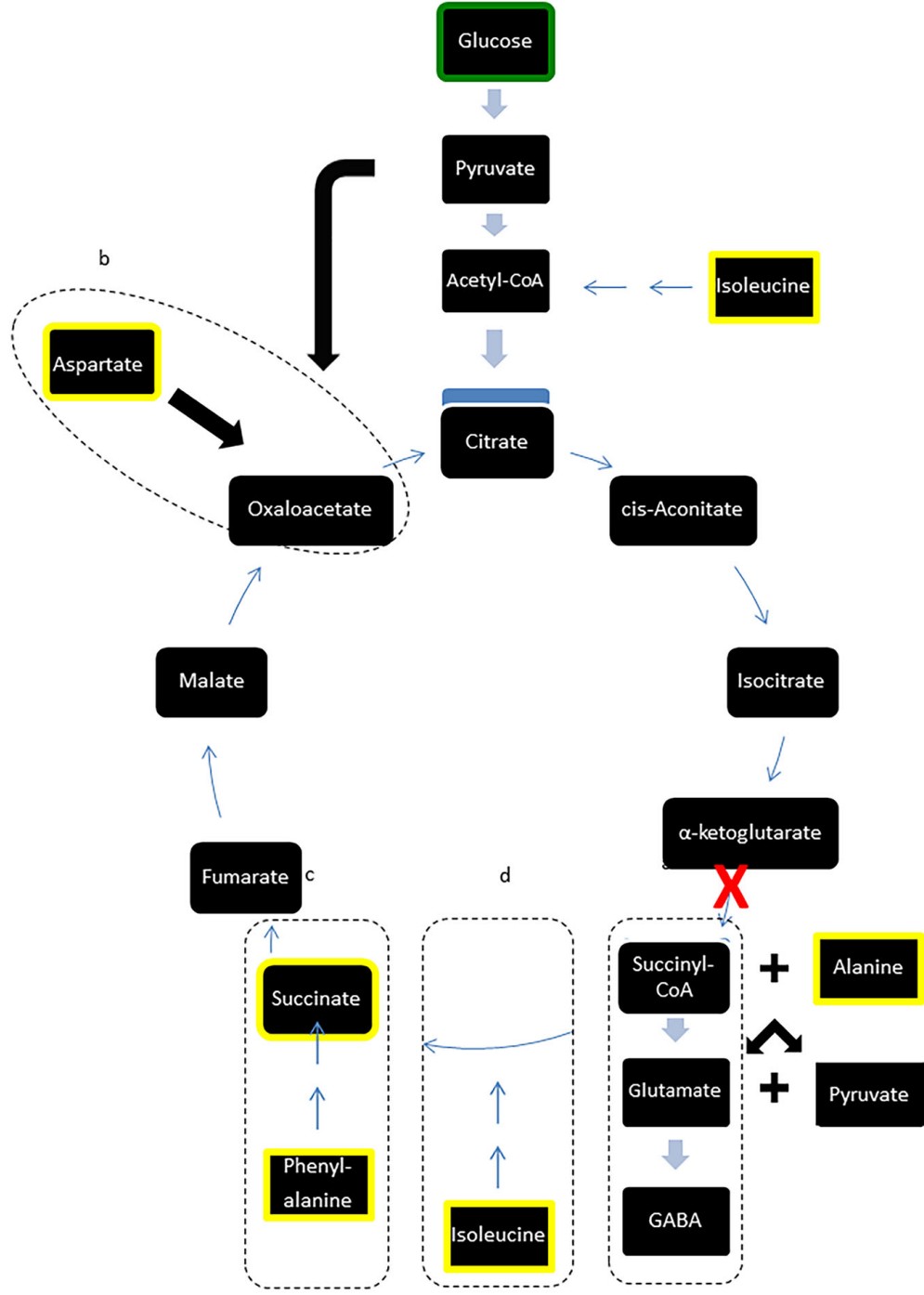

**Figure 3** **Modified TCA cycle showing effects of LI-rMS on metabolic profile of B50 cells.** Metabolites highlighted in yellow were significantly reduced in B50 cells following 1 and/or 10 Hz LI-rMS, compared to control B50 cells. The level of glucose (highlighted in green) was measured in our experiments and found to be unchanged by LI-rMS. The observed changes are proposed to result from increased *de novo* GABA synthesis (pathway a) that results in depletion of TCA cycle intermediates and precursors (pathways b, c, d). Suspected point of interference by LI-rMS is noted with ' *X* '.

of lipid rafts and cell membranes, but also modulates vesicle trafficking and exocytosis (*Churchward et al., 2005*; *Lang et al., 2001*; *Zhang et al., 2009*). A recent study in ageing mice found that rTMS reduced cholesterol that accumulated with age (*Wang et al., 2013*). This is important because cholesterol is associated with oxidative stress during normal aging (*Cutler et al., 2004*), and decreasing cholesterol levels can improve cognition in rats (*Wang et al., 2013*). A decrease of cholesterol following rTMS and LI-rTMS, as suggested by our data, may thus contribute to the cognitive improvements observed following rTMS treatment of Alzheimer's patients (*Hsu et al., 2015*), and also in healthy volunteers (*Miniussi & Ruzzoli, 2013*).

### Frequency-dependent metabolic changes

High and low frequency rTMS have different effects on neuronal circuits, with functional and genetic studies identifying distinct mechanisms in different neuronal cell types (*Grehl et al., 2015*; *Pell, Roth & Zangen, 2011*). Previous work using high intensity rTMS in anaesthetised cats demonstrated that metabolic changes were also frequency specific with increased glucose metabolism following high, but not low frequency stimulation (*Valero-Cabré, Payne & Pascual-Leone, 2007*). Our study provides a different approach, because we examined metabolic changes in a uniform population of neuronal cells with an inhibitory phenotype. Thus our finding that the effects of LI-rMS are stronger with 1 Hz than with 10 Hz matches the inhibitory effect of 1 Hz that is consistently reported in animal and human literature (*Chen et al., 1997*; *Maeda et al., 2000*; *Pell, Roth & Zangen, 2011*; *Trippe et al., 2009*). However, because LI-rTMS does not induce action potentials, it remains unclear whether the metabolic changes we report are a direct consequence of the electromagnetic field, or whether they are secondary to the changes in excitability induced by rTMS.

B50 neuroblastoma cells are derived from the rat central nervous system (*Schubert et al., 1974*) and have been used extensively to study the pathways involved in cell death, proliferation and migration (e.g., *Honma et al., 1996*). Although cells grown in culture remain a highly simplified model of the CNS (discussed in *Grehl et al., 2015*), B50 cells have been reported to fire regenerative action potentials, and contain high levels of the GABA bioythetic enzyme, glutamate decarboxylase (GAD), as well as GABA itself, suggesting that they retain some key functional features of inhibitory neurons (*Schubert et al., 1974*). Future studies in large, uniform populations of neuronal-like cells will contribute to our understanding not only of the mechanism whereby cells detect electromagnetic fields, but also of the signalling events such as kinase and phosphatase cascades, that underpin the complex cellular responses such as apoptosis, differentiation, migration and proliferation.

## CONCLUSION

Our study showed that LI-rMS induces a depletion of metabolites that suggested higher spontaneous activity may have resulted in an increase in GABA release. While the B50 cell line is composed of cells that are electrically excitable and homogenous thus allowing for a targeted examination of the effects of LI-rMS on a defined cell population, the absence of defined neural circuits and other cellular contributors (e.g., excitatory neurons

and glia) limits the degree to which our results can be extrapolated to the human brain. Nonetheless, the changes we describe provide novel insights into how LI-rTMS may modulate neural tissue and contribute to our understanding of the therapeutic application of electromagnetic brain stimulation.

## ACKNOWLEDGEMENTS

The authors are grateful to Dr Alex Tang for advice and assistance with magnetic field measurements, to Dr Kartik Iyer for feedback on the manuscript, and to Mrs Marissa Penrose-Menz for assisting with figure preparation.

### Funding

This work was supported by Murdoch University. At the time of the work, Ivan Hong was funded by an International Postgraduate Research Scholarship and Jennifer Rodger was a National Health and Research Council (Australia) Senior Research Fellow (APP1002258). The funders had no role in study design, data collection and analysis, decision to publish, or preparation of the manuscript.

### Grant Disclosures

The following grant information was disclosed by the authors:
Murdoch University.
International Postgraduate Research Scholarship.
National Health and Research Council (Australia) Senior Research Fellow: APP1002258.

### Competing Interests

Jennifer Rodger is an Academic Editor for PeerJ.

### Author Contributions

- Ivan Hong conceived and designed the experiments, performed the experiments, analyzed the data, prepared figures and/or tables, authored or reviewed drafts of the paper, approved the final draft.
- Andrew Garrett conceived and designed the experiments, performed the experiments, contributed reagents/materials/analysis tools, prepared figures and/or tables, authored or reviewed drafts of the paper, approved the final draft.
- Garth Maker conceived and designed the experiments, performed the experiments, analyzed the data, contributed reagents/materials/analysis tools, prepared figures and/or tables, authored or reviewed drafts of the paper, approved the final draft.
- Ian Mullaney, Jennifer Rodger and Sarah J. Etherington conceived and designed the experiments, contributed reagents/materials/analysis tools, prepared figures and/or tables, authored or reviewed drafts of the paper, approved the final draft.

### Data Availability

The raw data are provided as Data S1.

## Supplemental Information

Supplemental information for this article can be found online at http://dx.doi.org/10.7717/peerj.4501#supplemental-information.

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
