# Peer review of "Repetitive low intensity magnetic field stimulation in a neuronal cell line: a metabolomics study"

_PeerJ, doi:10.7717/peerj.4501_

## Round 0.1 · original submission · Minor Revisions

We have received the following two reviews of your paper with minor revisions suggested. Please address these comments.

Reviewer 1 ·

Basic reporting

Low intensity repetitive transcranial magnetic stimulation (LI-rTMS) was found recently as a beneficial way to treat depression, pain, and probably other neurological problems. The LI-rTMS is a non-invasive form of brain stimulation that induces structural and functional brain plasticity. However, the mechanisms by which this magnetic stimulus affect the brain, and the potential for LI-rTMS to promote axonal regeneration following neurotrauma has not been investigated yet. Here the authors used a rat neuroblastoma cells to follow the effect of LI-rTMS at a cellular and biochemical level. They found that LI-rMS reduces selected metabolites shortly after stimulation, without affecting the main energy supplies. The effects were frequency-specific with 1 Hz stimulation having stronger effects than 10 Hz. Overall, this is a well written and interesting study that provides novel insights into the modulation of neural tissue by LI-rMS. However, the scope of the article is somewhat limited, and suggestions how to expend it as well as a few minor comments are as follows:
1) The effect of LI-rTMS on metabolites is well presented, and the findings are important and deserve publication. However, in my mind, the system of B50 cells has a much wider potential in studying the mechanisms by which the magnetic field is sensed by the cells (what is the antennae that absorbs it), what are the signaling moieties involved (AMPK? mTOR? Calcium?) and what other physiological effects (e.g. morphology, proliferation, migration) are influenced. If the authors feel that these points are out of the scope of this article, they should at least discuss them under “Discussion”.
2) The logic of Figure 2 is not clear to me, and should be better explained. Can the authors specify the metabolites measured in each point?
3) In figure 3, it would be beneficial to add where is the suspected point of LI-rTMS interference. In addition, it is suggested to increase the font, so that the names of the metabolites become clearer.

Experimental design

The authors used a rat neuroblastoma cells to follow the effect of LI-rTMS at a cellular and biochemical level.
I have no problem with this design.

Validity of the findings

The results seem to be valid.

Additional comments

1) The effect of LI-rTMS on metabolites is well presented, and the findings are important and deserve publication. However, in my mind, the system of B50 cells has a much wider potential in studying the mechanisms by which the magnetic field is sensed by the cells (what is the antennae that absorbs it), what are the signaling moieties involved (AMPK? mTOR? Calcium?) and what other physiological effects (e.g. morphology, proliferation, migration) are influenced. If the authors feel that these points are out of the scope of this article, they should at least discuss them under “Discussion”.
2) The logic of Figure 2 is not clear to me, and should be better explained. Can the authors specify the metabolites measured in each point?
3) In figure 3, it would be beneficial to add where is the suspected point of LI-rTMS interference. In addition, it is suggested to increase the font, so that the names of the metabolites become clearer.

·

Basic reporting

Paper adheres to all basic reporting requirements.

Minor note: references need to be rechecked
Eg: some paper titles all start with capitals, others do not. Some journal titles are given in abbreviated form (line 308). Check page numbering (line 393)

Experimental design

1) The analytical method of choice – GC-MS – is a highly sensitive platform, able to produce 100’s of identifiable peaks. From this study: “Based on GC-MS profiling, a total of 18 reproducible intracellular metabolites were identified from the total ion chromatograms (TIC)”. Why were only 18 metabolites identified? On what basis is ‘reproducible’ used/defined as here? Were there data reduction steps taken? More description of the data-processing is required.

2) In the methods: “The supernatant was evaporated before adding 500 μl of water and freeze-drying again.” – what was the point of adding the water and freeze-drying again? Especially immediately before the derivatization steps as any trace of water reverses derivatization.

Validity of the findings

1) The hypothesis generated from this study is built around the following statement: “observed depletion of metabolites was consistent with an increase in GABA release as a result of higher spontaneous activity”. Yet, GABA levels are not quantitatively shown/given in the paper – the statement that GABA increases needs to be backed up by empirical proof.

2) The abstract is inconsistent with the main text. The groups in the abstract are stated as: “We delivered LI-rMS (10 mT) at 1 Hz and 10 Hz (n=5 wells per group)”. In the legend for Figure 2 & Table 1, the groups are stated as: “cells stimulated with LI-rMS at 1 Hz (n=4), 10 Hz (n=6) and unstimulated controls (n=6)”. Number of wells per group is not given in the main text.

3) Regarding number of wells per group – six, let alone four (as given for 1Hz in figure 3 and table 1), is a very low number (n) to be using for statistical analysis. A main pillar behind basic statistics is that a large(r) sample size will give more reliable results – an indulgence not often affordable in a typical metabolomics study. Should the study conduct research with an extremely small sample size (n ≤ 5), the t-test can be applied, as long as the effect size is expected to be large [see: De Winter, J.C., 2013. Using the Student's t-test with extremely small sample sizes. Practical Assessment, Research & Evaluation, 18(10)]. Often the Cohen’s d effect size is used, with cut off values of small (d = 0.2), medium (d = 0.5) and large (d ≥ 0.8) [C. Carson, The effective use of effect size indices in institutional research. Citováno dne, 11, 2016]. Hence, for a small sample size, a low p-value (p<0.05) combined with a high d-value (d>0.8) is recommended as an indicator of being univariately significant.

4) In line with the comment above – what are the cut-off values for the F-statistic given in Table 1? No mention of the F-statistic is described in the text (reporting, results, nor discussion), and why was this statistic (not commonly used in metabolomics) used as a statistical measure for metabolite selection? Similarly, what is the cut-off value for significance with regard to the fold change? GC-MS is quantitative – means, standard deviations and ranges would add value to Table 1? This, and the previous comment, is aimed at helping to improve upon Table 1 and the reporting of the quantitative univariate measures from the metabolomics data.


Minor note: oxalosuccinate is a highly unstable intermediate that immediately undergoes decarboxylation to 2-ketoglutarate. Hence, oxalosuccinate is not typically shown as an intermediate in the TCA cycle. Oxalosuccinate is not important in this study; hence it seems out of place in Figure 3.

Additional comments

This is a well-designed study that examines the effects of electromagnetic stimulation of brain cells, specifically neuroblastoma cells. The therapeutic applications of electromagnetic stimulation of the brain holds much potential. Studies such as this will help expand our understanding of how and what is occurring at the metabolic level. While adequately covering the need for such a study, the reporting of the metabolomics aspect here still needs to be improved upon (see other comments).

---

## Round 0.2 · accepted · Accept

Dear Jenny,

Thanks for re-submitting the manuscript. I am writing to inform you that your manuscript - Repetitive low intensity magnetic field stimulation in a neuronal cell line: A metabolomics study - has been Accepted for publication. Congratulations!